# Independent and Joint Associations of Physical Activity and Dietary Behavior with Older Adults’ Lower Limb Strength

**DOI:** 10.3390/nu12020443

**Published:** 2020-02-10

**Authors:** Ting-Fu Lai, Chien-Yu Lin, Chien-Chih Chou, Wan-Chi Huang, Ming-Chun Hsueh, Jong-Hwan Park, Yung Liao

**Affiliations:** 1Department of Health Promotion and Health Education, National Taiwan Normal University, 162, Heping East Road Section 1, Taipei 106, Taiwanliaoyung@ntnu.edu.tw (Y.L.); 2Graduate School of Sport Sciences, Waseda University, 2-579-15 Mikajima, Tokorozawa City 359-1192, Japan; chienyulin@akane.waseda.jp; 3Graduate Institute of Sport Pedagogy, University of Taipei, No. 101, Sec. 2, Jhongcheng Rd., Shilin Dist., Taipei 11153, Taiwan; 4Health Convergence Medicine Laboratory, Biomedical Research Institute, Pusan National University Hospital, 179, Gudeok-Ro, Seo-Gu, Busan 49241, Korea

**Keywords:** older adult, physical function, dietary behavior, accelerometer, physical activity

## Abstract

Studies have indicated that sufficient physical activity levels and balanced dietary behavior are independently related to physical function in older populations; however, their joint association with physical function remain unclear. This study examined the independent and combined associations of sufficient physical activity and balanced selection of foods with lower limb strength among 122 older Taiwanese adults living in community (mean age: 69.9 ± 5.0 years). The assessments included accelerometer-measured moderate-to-vigorous physical activity (MVPA) and self-reported selection of foods. Lower limb strength performance was measured using the five times sit-to-stand test. Binary logistic regression analyses were performed to estimate the associations in question before and after adjusting for potential confounders. The results showed that in the adjusted model, lower limb strength had no significant independent association with either meeting the recommended level of MVPA or balanced selection of foods. Compared to older adults who neither met the recommended MVPA level nor reported a balanced selection of foods, those who conformed to both these criteria were more likely to have better lower limb strength (odds ratio = 6.28, 95% confidence interval = 1.36–29.01) after adjusting for covariates. Health promotion initiatives addressing disability prevention for older adults need to consider promoting both MVPA and food selection.

## 1. Introduction

Owing to prominent advances in healthcare, the average life expectancy of the global population has been on the rise, and aging is now a global trend [1]. The proportion of older adults aged 60 years and above worldwide is rising; in the period from 2017 to 2050, this number is projected to more than double, growing faster than all other age groups [2]. Geriatric syndromes include functional decline, delirium, frailty, sarcopenia, pressure ulcers, and urinary incontinence are being increasingly prioritized in preventive healthcare [3], and with good reason; they are associated with adverse health outcomes including falls, institutionalization, disability, or even death among older adults [4,5]. Given its high prevalence among older populations, frailty is a major contributor to the high burden of these geriatric syndromes [6]. Frailty is characterized by an age-associated increase in vulnerability and decrease in the capability to recover physiological homeostasis after a destabilizing event [6]. Studies have shown a number of proximity indicators of deficiencies in functional physical performance (e.g., lower levels of hemoglobin and antioxidant capacity) and lower levels of lower limb strength that are related to age and increased presence of signs of frailty [7,8,9]. In addition, studies have indicated that better lower limb strength is associated with better quality of life [10] and lower risks of disability [11], falls [12], and type 2 diabetes [13]. While frailty is a dynamic process, the worsening of the condition is more common than improvement [14]. Therefore, it is crucial to identify influential factors, especially modifiable lifestyle behaviors, and develop effective strategies to shield older adults from frailty.

Some energy balance-related behaviors such as sufficient physical activity levels and balanced dietary behavior play key roles in building and maintaining lower extremity muscle strength via muscle cell formation and sufficient nutrient intake (e.g., protein) [15]. According to international recommendations, an adequate amount of regularly performed exercise could reduce the risk of mobility limitation in older populations [16]. A previous study with community-dwelling older adults showed positive associations between objectively measured physical activity and leg strength [17]. A systematic review of observational studies on dietary behavior showed that older adults who took higher protein in were associated with better lower limb performance than who took a lower volume of protein [18]. Furthermore, another study from Australia indicated that high consumption of a plant-based diet, regarded as a healthy dietary pattern, is positively associated with lower limb muscle strength among middle-aged women [19]. Sufficient levels of physical activity and balanced dietary behavior could function to enhance older adults’ lower extremity muscle strength independently and further reduce the risk of falls. However, limited studies have examined the combined associations of recommended physical activity levels and balanced dietary behavior with lower limb strength in older populations. Therefore, whether there is an interaction between sufficient physical activity and balanced dietary behavior as they relate to lower limb strength remains unclear.

This study was aimed at gaining an understanding of the independent and joint associations of sufficient physical activity levels and balanced dietary behavior (i.e., a habit of consuming a balanced selection of foods) with older adults’ lower limb strength. In this study, it was hypothesized that: Thirty minutes of moderate-to-vigorous physical activity (MVPA) per day would be independently associated with better lower limb strength.A balanced selection of foods based on recommendations would be independently associated with better lower limb strength.A combination of 30 minutes of MVPA per day and a balanced selection of foods would be more strongly associated with better lower limb strength than either factor alone.

## 2. Materials and Methods

### 2.1. Participants

We collected cross-sectional data on community-dwelling older adults (aged ≥ 65 years) living in Taipei City, Taiwan, who had the ability to walk independently without using an assistive device. The recruitment was announced through local advertisements and notifications at health centers, and those interested in the study could contact the recruiters easily. Detailed information regarding the recruitment process has been presented elsewhere [20]. Physical performance, such as lower limb strength, was tested on-site by the interviewers. After the on-site examination, the interviewers gave each participant an accelerometer, which was to be worn on their waist for a whole week. Each participant who completed the questionnaire and on-site tests and returned the accelerometer after seven consecutive days would receive a convenient store voucher (USD 7) as a reward. One hundred and seventy participants were initially enrolled. After excluding those who did not wear the accelerometer with enough valid time (n = 22) and those who wore the accelerometer with enough valid time but had missing or incomplete data (n = 26), the data of 122 participants were analyzed.

### 2.2. Research Ethics

We obtained ethical approval from the Research Ethics Committee of the National Taiwan Normal University (REC number: 201711HM003). Informed consent was obtained prior written from each participant.

### 2.3. Lower Limb Strength

We assessed functional lower limb strength using the five times sit-to-stand (STS) test, which is considered to be a valid and reliable measure for older adults [21] that has been widely utilized in previous studies [22,23]. Previous studies showed the retest reliability (intraclass correlation coefficient) ranged from 0.89 to 0.96 [24,25]. Participants were provided with standardized instructions for the five times STS test. Then, the time taken to perform five repetitions of the STS maneuver as fast as possible was recorded. Lower limb strength performance was categorized into two groups with good and bad performance using sex-specific median (men: 6.95 seconds; women: 6.88 seconds) of the five times STS test. 

### 2.4. Objectively Measured Physical Activity 

We used the ActiGraph wGT3X-BT (ActiGraph, LLC, Pensacola, FL, USA) to assess steps taken as well as the time spent on physical activity (≥100 counts/min) and sedentary behavior (<100 counts/min). The triaxial ActiGraph wGT3X-BT has been proved high intra-monitor reliability and validated with acceptable criteria [26]. The participants wore the accelerometer to record their activities for one consecutive week (all five weekdays and both weekend days). According to the physical activity guidelines of the American College of Sports Medicine, in the context of older adults, the accumulation of 30 minutes of MVPA per day is associated with health benefits such as functional ability, mental and cognitive health, and prevention of some chronic conditions [27]. Therefore, we extracted the MVPA data (≥2020 counts/minute [28]) and divided them into two groups based on 30 minutes of MVPA per day [27]. Intervals of MVPA sustaining at least ten successive minutes with a one-minute quota below the MVPA threshold were identified as MVPA bouts and included in the analysis [26]. In this study, accelerometer data of participants with at least four valid days including one day of weekend were analyzed [29]. A period of 10 hours (600 minutes) or more of monitor wear time was classified as a valid day. We downloaded accelerometer data using ActiLife computer software (version 6.0, Pensacola, FL, USA). 

### 2.5. Self-Reported Selection of Foods

We asked participants to report yes or no to the question, “Do you have a habit of ensuring a balanced intake of food groups based on the Taiwanese dietary guidelines?” The instruction “for example, three servings of vegetables and two servings of fruit every day” according to Taiwanese dietary guidelines [30] was provided along with the question. This question represented the dietary behavior of consuming a balanced selection of foods, which was used widely in previous studies in relation to older adults in Taiwan [20,31,32]. 

### 2.6. Covariates

We used interviewer-administered questionnaires to assess the covariates, including sociodemographic characteristics (e.g., age, sex, marital status, living status, and educational level), health status (body mass index (BMI), general health, depression, and chronic diseases), and sedentary behavior based on previous findings [33,34]. We categorized participants’ sociodemographic variables into two groups as follows: 65–74 or ≥75 years for age group; married or unmarried for marital status; living with others or living alone for living status; and university degree or lower for educational level. The self-rated health status was assessed by a five-point Likert scale, ranging from (1) “very good” to (5) “very bad.” Responses of at least three points were classified as “good,” and the others were classified as “bad.” We also requested the participants to report whether they felt depressive moods frequently (yes or no) and received diagnoses and/or took medicine for three of the most common chronic diseases in Taiwan, namely, hypertension, hyperlipidemia, and diabetes (yes or no); the corresponding prevalence rates were 53.5%, 26.8%, and 22.6% [35]. We calculated BMI as self-reported weight in kilograms divided by the square of the self-reported height in meters, in accordance with the definition provided by the Ministry of Health and Welfare of Taiwan. The participants were categorized as “normal (18.5–24 kg/m^2^)” or “overweight (>24 kg/m^2^)” [30]. Accelerometer-measured sedentary time (≤99 counts/minute [28]) was included as a categorical variable. Based on studies that have indicated that over 9 hours/day of sedentary time is associated with increased mortality risks, the variable was divided into “≥9 hours/day” and “<9 hours/day” [36]. We also adjusted for accelerometer wear time for analyses in relation to physical activity. 

### 2.7. Statistical Analyses

We examined the degree of multicollinearity between the studied variables by checking their variance inflation factors (VIFs), with a value higher than 10 indicating a high level of multicollinearity [36]. In sensitivity analyses, we stepwise omitted variables with the highest VIF from the regression models to confirm the robustness of our results, in consideration of our small sample. Nagelkerke’s R^2^ was calculated to evaluate the explanatory power of different models [37,38]. Binary logistic regression analyses were performed to examine the independent and joint associations of sufficient MVPA level and balanced selection of foods with lower limb strength. The odds ratios (ORs) and 95% confidence intervals (CIs) were calculated for these associations. In the analysis of the joint association, the group that neither met the recommended MVPA level (30 minutes/day) nor reported a balanced selection of foods was regarded as the reference group. The aforementioned covariates were adjusted for in the adjusted models. Statistical analyses were conducted using SPSS 24.0 (IBM Corp., Armonk, NY, USA); the level of significance was set at *p* < 0.05.

## 3. Results

Table 1 depicts the sociodemographic characteristics, health status, and health behaviors of 122 participants with valid and completed data for the analysis. The mean accelerometer wear time was 15.4 hours per day. The mean age was 69.9 ± 5.0 years; 18.0% of the sample aged 75 years and above. Of the participants, most were women (71.3%), were married (66.4%), lived with others (90.2%), had a low educational level (77.9%), had a normal BMI (51.6%), self-rated their health status as poor (68.9%), did not report frequent depressive moods (86.9%), did not have chronic diseases (60.7% for hypertension; 70.5% for hyperlipidemia; and 81.1% for diabetes), sat for at least nine hours per day (81.1%), did not engage in sufficient MVPA (69.7%), and reported a balanced selection of foods (73.0%). 

Table 2 shows the independent associations of accelerometer-measured physical activity and self-reported selection of foods with lower limb strength in older adults. In the unadjusted model, older adults who met the recommended level of MVPA (OR = 2.72, 95% CI = 1.18–6.30) or reported a balanced selection of foods (OR = 2.97, 95% CI = 1.25–7.04) were more likely to have better lower limb strength. However, the associations of meeting the recommended MVPA level (OR = 2.14, 95% CI = 0.79–5.79) and reporting a balanced selection of foods (OR = 2.48, 95% CI = 0.91–6.74) were attenuated after adjusting for covariates. In the sensitivity analyses of sufficient MVPA (Appendix A
Table A1) and balanced selection of foods (Appendix A
Table A2), in which variables showing the highest VIF were omitted stepwise, the results across different models were all similar. The final model with the largest explanatory power was regarded as the main one for analysis. 

Table 3 depicts the joint association of accelerometer-measured physical activity and self-reported dietary behavior with lower limb strength in older adults. In the unadjusted model, compared to the older adults who neither met the recommended level of MVPA nor reported a balanced selection of foods, those who met both criteria were more likely to have better lower limb strength (OR = 8.00, 95% CI = 2.25–28.48). By contrast, no significant associations were found in other groups. Similar associations were found in the adjusted models; after adjusting for covariates, there was an association with lower limb strength (OR = 6.28, 95% CI = 1.36–29.01) in only those who met the recommended level of MVPA and reported a balanced selection of foods.

## 4. Discussion

Our data showed no independent associations of achieving the recommended level of MVPA and balanced selection of foods with lower limb strength among older adults. However, a joint association was observed. Older adults who engaged in the recommended level of MVPA and reported a balanced selection of foods exhibited six times better lower limb strength than those who neither met the recommended level of MVPA nor followed a balanced selection of foods.

To our knowledge, this analysis is the first examining the joint pattern of physical activity and dietary behavior of selecting foods in the context of older adults’ lower limb strength. To ensure objective measurement of MVPA, we used an accelerometer instead of a self-reported questionnaire. However, this study has several limitations that should be considered. First, the sample size of the participants was small, and the participants may not be representative of all Taiwanese older adults as those who were generally in good health may have been more likely than others to participate in this study. Our data showed that older people were less likely to complete our tests and thus were excluded from our analyses: the mean age of included (*n* = 122) and excluded (*n* = 26) participants were 70.0 and 74.1 years (*p* <0.001 for *t*-test), respectively. Second, these participants were not screened for cognitive ability with a standardized measure in the recruitment period. Cognitive impairment may interfere with their capability to respond to the questions or understand the testing procedure. Third, this study used a single question for assessing the general condition of dietary behavior, not taking into account specific dietary constituents such as proteins [16], amino acids, and vitamin D and E [39,40], which have been shown to be related to lower limb strength. This question may be unable to elaborate detailed information on individuals’ dietary behaviors. In addition, self-reported measurements of dietary behavior and some of the covariates may cause biases. Fourth, physical problems such as knee osteoarthritis and low back pain [41] were not considered as covariates in this study. However, we included the most common chronic diseases in our analyses to control the influence of physical problems to some extent. Future studies are suggested to account for these variables in the analyses. Finally, we cannot make causal inferences regarding the associations of physical activity and dietary behavior with lower limb strength because the study was cross-sectional design.

Based on our data, there were no independently significant associations of physical activity and dietary behavior with lower limb strength, which is inconsistent with previous findings [17]. A previous study with a larger sample size (*n* = 636) that used a dynamometer to measure lower limb strength showed a positive association with accelerometer-measured MVPA [17]. In comparison to previous studies, the present sample size (*n* = 122) was relatively small, which might have resulted in the statistical power being insufficient to identify an association. In addition, lower limb strength has been demonstrated to display a marked sex-based difference [42]. However, in this study, we found a similar performance in lower limb strength between men (mean time: 7.56 ± 2.19 seconds) and women (mean time: 7.44 ± 2.72 seconds), which was generally better than the performance showed in previous study [43]. The performance of lower limb strength with small variance across our samples and a general better performance than data for previous studies may have mitigated the associations. In previous studies, nutrient intake in older adults was related to better muscle mass and physical performance [44], which is inconsistent with our results. Studies on the topic have used different definitions of a balanced selection of foods. For example, there have been studies assessing nutritional status based on questions on dietary intake volume, neuropsychological problems, BMI [45], and specific nutrients [39,40] rather than the national government’s recommendations for general selection of foods, as done in this study. In the context of lower limb strength performance in older adults, our results may cast some doubt as to whether the general dietary behavior of selecting foods considered in this study reflects the same concept of nutritional status used in previous literature. 

Consistent with our expectation, older adults who achieved sufficient MVPA and reported a balanced selection of foods had better lower limb strength performance than those who only focused on MVPA or dietary behavior. This implies a potential interaction effect of physical activity and dietary behavior in the context of lower limb strength. Sufficient physical activity and a balanced diet are beneficial for building and maintaining muscle strength [15] and leave less time for engaging in unhealthy behaviors (i.e., sedentary behavior and pickiness about food) [46]. Alternatively, older adults with better lower limb strength performance tend to engage in more MVPA and thus consume a more nutritious diet.

In conclusion, our data showed a joint association, but not independent associations, of sufficient MVPA and balanced dietary behavior with lower limb strength in older adults. These findings have implications for health promotion initiatives to prevent disability in older adults, particularly those who engage in insufficient physical activity and consume an unbalanced diet. In addition, the interaction effect of physical activity and dietary behavior should be valued. In those who engage in sufficient physical activity but display unbalanced dietary behavior, resources targeted at strengthening the awareness of balanced dietary behavior may improve lower limb strength and vice versa. Future research into dietary behavior and lower limb strength considering different definitions of balanced diet in the context of older adults is needed.

## Figures and Tables

**Table 1 nutrients-12-00443-t001:** Characteristics of the participants.

Variables	Mean ± SD	Categories	*n*	%
Age (years)	69.9 ± 5.0	65–74	100	82.0%
		≥75	22	18.0%
Sex	-	Men	35	28.7%
		Women	87	71.3%
Marital status	-	Married	81	66.4%
		Unmarried	41	33.6%
Living status	-	Living with others	110	90.2%
		Living alone	12	9.8%
Educational level	-	University	27	22.1%
		Lower than university	95	77.9%
BMI (kg/m^2^)	24.2 ± 3.4	Normal	63	51.6%
		Overweight	59	48.4%
Self-rated health	-	Good	38	31.1%
		Poor	84	68.9%
Depressive symptoms		Yes	16	13.1%
		No	106	86.9%
Hypertension		Yes	48	39.3%
		No	74	60.7%
Hyperlipidemia		Yes	36	29.5%
		No	86	70.5%
Diabetes		Yes	23	18.9%
		No	99	81.1%
Sedentary behavior (hours/day)	10.1 ± 1.2	≥9	99	81.1%
		<9	23	18.9%
MVPA (minutes/day)	24.6 ± 23.2	Sufficient	37	30.3%
		Insufficient	85	69.7%
Balanced selection of foods	-	Balanced	89	73.0%
		Unbalanced	33	27.0%

SD: standard deviation; BMI: body mass index; MVPA: moderate-to-vigorous physical activity.

**Table 2 nutrients-12-00443-t002:** Independent associations of objectively measured physical activity and self-reported dietary behavior with lower limb strength in older adults.

	Unadjusted		Adjusted ^a^	
	OR (95% CI)	*p*-Value	OR (95% CI)	*p*-Value
Physical activity				
Not engaging in 30-min MVPA per day	1.00		1.00	
Engaging in 30-min MVPA per day	2.72 (1.18–6.30)	0.02 *	2.14 (0.79–5.79)	0.14
Dietary behavior				
Not meeting dietary guidelines	1.00		1.00	
Meeting dietary guidelines	2.97 (1.25–7.04)	0.01 *	2.48 (0.91–6.74)	0.08

OR: odds ratio; CI: confidence interval; MVPA: moderate-to-vigorous physical activity; * *p* < 0.05. ^a^ Adjusted for age group, sex, marital status, living status, educational level, body mass index, self-rated health, depressive symptoms, hypertension, hyperlipidemia, diabetes, sedentary behavior, and further adjusted for accelerometer wear time for physical activity only.

**Table 3 nutrients-12-00443-t003:** Combined association of objectively measured physical activity and self-reported dietary behavior with lower limb strength in older adults.

	Unadjusted		Adjusted ^a^	
	OR (95% CI)	*p*-Value	OR (95% CI)	*p*-Value
Not meeting MVPA guidelines × Unbalanced selection of foods (n = 23)	1.00 (ref.)	-	1.00 (ref.)	-
Not meeting MVPA guidelines × Balanced selection of foods (n = 62)	2.29 (0.83–6.33)	0.11	2.02 (0.62–6.62)	0.24
Meeting MVPA guidelines × Unbalanced selection of foods (n = 10)	1.52 (0.33–7.15)	0.59	1.19 (0.19–7.59)	0.86
Meeting MVPA guidelines × Balanced selection of foods (n = 27)	8.00 (2.25–28.48)	0.01 *	6.28 (1.36–29.01)	0.02 *

OR: odds ratio; CI: confidence interval; MVPA: moderate-to-vigorous physical activity; ** p* < 0.05. ^a^ Adjusted for age group, sex, marital status, living status, educational level, body mass index, self-rated health, depressive symptoms, hypertension, hyperlipidemia, diabetes, sedentary behavior, and accelerometer wear time.

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
