# Peer review of "Independent and Joint Associations of Physical Activity and Dietary Behavior with Older Adults’ Lower Limb Strength"

_nutrients, 2020, doi:10.3390/nu12020443_

Round 1

Reviewer 1 Report

Title and aim is clear, title is informative and relevant Regarding balanced dietary behavior, authors need to provide enough information in stead of referring to other articles, such as dietary recall time, brief description of balanced diet. Authors asked only one question to assess the participants’ dietary habit. It could be one of the main weak points in this study. At least, authors should have mentioned what’s the question in exact word? Also, authors stated that “An example of balanced dietary behavior according to Taiwanese dietary guidelines [33] was provided accompanied with the question”. This sentence is not clear. Authors did not mention about cognitive impairement of the participants. More than 20% of those aged 60 years or more have a neurological disorder (WHO Mental Health of Older Adults, 2017). Did the authors assess for cognitive impairment before inclusion of participants? Cognitive impairment could interfere with understanding of the questions, testing procedures or may affect ActiGraph wearing time. Author should clearly mention how they assess participants´ self-reported health status? Author did not mention how they selected potential confounders. Such as, employment (full-time job or not). All participants were 65 years or older. It is less likely for them to remain employed at that age. The authors did not mention pertinent limitations of their study. Self-reported dietary behavior is a potential source of bias; which they could have acknowledged. Overall, the data are somewhat difficult to interpret, partially because the authors are using misleading covariates as mentioned before.

Reviewer 2 Report

This study examined the independent and combined associations of sufficient physical activity and balanced diet with lower limb strength in the old age. Previous studies have associated better lower limb strength with health. In fact, in my University there are two very powerful research groups studying the effect of nutrition on the one hand and physical activity  (Physical Activity Research Group, PROFITH), on the other, on various parameters related to healthy aging.

However, the novelty of this studio is examining the joint patterns of physical activity and dietary behavior. It would have been interesting to have made a more complete food consumption record to be able to relate it to the intake of a particular nutrient and not only with an adequate nutritional profile. Although the impact of the results is modest due to the limitations of the study, as the authors themselves state, the study is very well done and very interesting.

Author Response

Dear Prof. Lluis Serra-Majem & Prof. Maria Luz Fernandez,

Editor-in-Chief

Nutrients

Re: Independent and Joint Associations of Physical Activity and Dietary Behavior with Older Adults’ Lower Limb Strength (Manuscript ID: nutrients-689118)

The authors thank the reviewers for reading our manuscript so thoroughly and providing such constructive feedback. The quality of our manuscript has certainly improved as a result of these comments. We have listed out our responses to the reviewers’ comments point by point and corresponding changes in our revised paper in the below. To incorporate revisions in response to reviewers’ comments, we have made changes in the main text, tables, and appendixes. The revised and new sentences are highlighted in red font in the revised manuscript.

We hope that you will find these adjustments satisfactory and that the revised version will be acceptable for publication in special issue of "Diet, Lifestyle and Healthy Ageing" in Nutrients.

Yours sincerely,

Ming-Chun Hsueh and Jong-Hwan Park

Corresponding authors

Responses to the Reviewer 2:

Comments to the Author

This study examined the independent and combined associations of sufficient physical activity and balanced diet with lower limb strength in the old age. Previous studies have associated better lower limb strength with health. In fact, in my University there are two very powerful research groups studying the effect of nutrition on the one hand and physical activity (Physical Activity Research Group, PROFITH), on the other, on various parameters related to healthy aging. However, the novelty of this studio is examining the joint patterns of physical activity and dietary behavior. It would have been interesting to have made a more complete food consumption record to be able to relate it to the intake of a particular nutrient and not only with an adequate nutritional profile. Although the impact of the results is modest due to the limitations of the study, as the authors themselves state, the study is very well done and very interesting.

Response: Thank you very much for your comment.

Reviewer 3 Report

The present cross-sectional study investigated whether the adherence to either physical activity or dietary guidelines or both is related to lower limb strength in community-dwelling older people. The topic is of interest regarding preventive strategies and the paper is clearly structured. Weaknesses are seen in the small sample size (n=122) and the used statistical approach. Please find my comments point by point below.

Introduction:

L44-45: Please give a reference for the definition of frailty.

L46-47: It is stated “In studies measuring lower limb strength using either biomarkers or physical function tests, lower levels of lower limb strength were related to advanced in years and increased presence of signs of frailty.” I was wondering how strength can be measured by biomarkers, please clarify this. Moreover, the meaning of “were related to advanced in years” is unclear.

L54-55: It is stated, “Some energy balance-related behaviors such as sufficient physical activity levels and balanced dietary behavior play key roles in building and maintaining lower extremity muscle strength via muscle cell formation and sufficient nutrient intake (e.g., protein and high-density lipoprotein).” Please explain the meaning of high-density lipoprotein in relation to sufficient nutrient intake. In my understanding HDL is a substance build in the body but not a nutrient requiring sufficient intake.

L61-63: Please be precise with your wording. In the mentioned meta-analysis by Trevisan et al. nutrient intake was not included as variable of interest. The work focused on nutritional status and BMI in relation to falls.

Please define the term balanced dietary behavior in context to your research question, as dietary behavior is an umbrella term covering aspects of food, nutrient and energy intake. The term “balanced” can consequently refer to energy balance, to sufficient nutrient intake or to a balanced selection of foods.

Methods:

L79 ff.: The study used an interviewer-administrated questionnaire to collect data. Were people screened for cognitive impairment before conducting the interviews to avoid misreporting? Please describe how cognitive status was considered in the study.

L83: It is stated “Detailed information regarding the recruitment process has been presented elsewhere [20].” Please describe just briefly, how participants were recruited – e.g. by using advertisements or data of the registration offices. Was a sample size calculation conducted or were any stratification criteria used, e.g. gender, level of physical activity, adherence to dietary guidelines, to have sufficient group sizes to answer the research question?

L89-90: Forty-eight of the 170 enrolled participants were excluded due to missing data. A comparison of the participants’ characteristics between in- and excluded participants would give information about a potential selection bias.

L101-102: To differentiate between participants with normal and reduced lower limb strength median split was used. Cut-off values for men and women were 6.95 and 6.88 seconds, respectively. Cut-off values described in the literature e.g. by Cruz-Jentoft et al. (Age Aeging 2018) or Guralink et al. (J Gerontol A Biol Sci Med Sci) are distinctly higher (15s and 11s) indicating that the study sample of the present study was rather fit. This might also be a reason for the lack of association between physical activity or dietary behavior and lower limb strength and might be discussed.

L120-125: The assessment of dietary behavior is described. However, for the reader is it not clear how the adherence to a balanced diet was defined. Please mention the six categories of nutrients, which were asked for, in the text and add if the answer categories were just answered with yes or no or whether a more detailed scoring system e.g. a 5-point Likert scale was used. Moreover, it should be added how the groups (balanced/unbalanced) were build. Did the participants need to fulfill all six categories? Was the used scale for dietary behavior validated against another measure of dietary intake e.g. dietary records, 24h-recall, food frequency questionnaire?

L126ff: In the results, self-rated health is mentioned as additional variable not described in the methods. Please add how the variable was assessed. Regarding physical activity, dietary behavior and lower limb strength further measures of physical and psychological health e.g. multimorbidity, polypharmacy, pulmonary diseases or depressive symptoms might be of interest to describe the sample. Were these variables assessed in the study?

L127-128: “Self-reported sociodemographic characteristics, health behaviors, and health status were  assessed via interviewer-administered questionnaires.” The sentence is a repetition of L83-84.

L130-133: Were body weight and height objectively measured or assessed by self-reports? Please add the type of measurement to the description as self-reports often underestimate body weight and overestimate body height.

L134-135: Sedentary time was used as a covariate. Could you please describe which data were divided by nine hours and why this value (9 hours) was used.

L137: To analyze the association between physical activity/dietary behavior and lower limb strength logistic regression was used and adjusted for potential confounders. Could you please describe on which basis the confounders were chosen for the analyses e.g. based on literature search or based on prior analyses. As the sample size in this study is rather small (n=122), the number of included covariates seems to be high (n=10). In the results distinct shifts in the estimates before and after adjustment are visible. Was multicollinearity of the variables tested before including them into the regression model? Did you check the model fit of your regression analyses?

Results

Table 2: The OR of meeting dietary guidelines turns after adjustment from 2.97 (1.25-7.04) to 0.44 (0.18-1.12). Is there an explanation why this had happened? Did you check which of the covariates is responsible for this shift?

Table 3: Please provide group numbers for the four groups presented in the table. It is assumed that the group meeting both MVPA guidelines and balanced dietary behavior is small as the OR is high and has a huge 95% CI.

Discussion

L189ff: With regard to the statistical approach the small sample size should be acknowledged as limitation of the study.

L201-205: “Based on our data, there were no independently significant associations of physical activity and dietary behavior with lower limb strength, which is inconsistent with previous findings [37]… A previous study with a larger sample size (N = 636) that used a dynamometer to measure lower limb strength showed a positive association with accelerometer-measured MVPA [37].” Please check whether the references are correct. The cited paper focusses on a consensus process for a set of physical performance measures in people with hip or knee osteoarthritis.

L203: It is stated “However, the association between MVPA and lower limb strength displayed a positive trend.” In the results an OR of 2.07 (0.79-5.43) and a p-value of 0.14 is described (Table 2). Based on the 95% CI and the p-value, this trend is not visible to me.

L212: Reference 39 is an overview paper and not a study as mentioned in the text.

L237: In the conclusion, a reduction in fall risk is mentioned. However, this aspect was not investigated in this study.

To improve readability of the manuscript a linguistic revision by a native speaker is recommended.

References

Ref 16: What is meant by “Society, A.G., et al.”?

Round 2

Reviewer 3 Report

Thank you for considering my comments and suggestions. The manuscript has been revised adequately. I have just few minor aspects.

The reference list does not include numbers, therefore it is not possible to connect the numbers given in the text with the respective references L20 and L213: “balanced diet of a selection of foods” In my view “diet of” can be deleted. Appendix Table 2: Column Adjusted for 10 variables: The confidence interval seems to be incorrect. 2.44 (9.31–6.41)
